# Preparation of Anticorrosive Epoxy Nanocomposite Coating Modified by Polyethyleneimine Nano-Alumina

Xin Liang [1], Cheng Hua [1], Mingrui Zhang [1], Yu Zheng [1], Shijie Song [2], Meng Cai [2], Yu Huang [2], Can He [2] and Xiaoqiang Fan [2],*

1  AVIC Chengdu Aircraft Industrial (Group) Co., Ltd., Chengdu 610073, China
2  Key Laboratory of Advanced Technologies of Materials (Ministry of Education),
   School of Materials Science and Engineering, Southwest Jiaotong University, Chengdu 610031, China
*  Correspondence: fxq@home.swjtu.edu.cn

**Abstract:** Aluminum alloys with low density and high specific strength have been widely used in marine engineering. Epoxy coatings, a simple and economical protection strategy, have been applied on alloy surfaces to prolong service life. However, a pure epoxy coating cannot provide long-term protection for metals in the marine environment. Hence, in this work, nano-alumina (nano-$Al_2O_3$) modified by polyethyleneimine (PEI) were added into epoxy coatings to enhance anticorrosion properties. Using Fourier transform infrared spectroscopy (FTIR), we found that the molecular chain of PEI was successfully grafted on the surface of nano-$Al_2O_3$, and the cross profile of coatings indicated that the modified nano-$Al_2O_3$ uniformly dispersed in the epoxy coating. Electrochemical impedance spectroscopy (EIS) results demonstrate that the coating resistance of the modified epoxy nanocomposite coating was 10 times higher than that of the pure epoxy coating after 3 days of immersion in 3.5 wt.% NaCl solution. Meanwhile, the surface morphologies and EDS-mapping of substrates after EIS testing show that the substrate coated with modified epoxy nanocomposite coating had the smallest amount of corrosion products. These results show that this modified epoxy nanocomposite coating has excellent anticorrosion performance.

**Keywords:** epoxy coating; nano-$Al_2O_3$; functionalization; corrosion resistance; electrochemistry





## 1. Introduction

Aluminum and its alloys have been widely applied in various fields, such as aerospace, automobile, and ocean engineering, by virtue of their low density and high stiffness–weight ratios [1,2]. However, aluminum alloys are susceptible to aggressive media in most corrosive environments, especially in the marine environment, resulting in huge economic losses and safety problems [3,4]. Thus, various strategies have been adopted to retard the corrosion rate of alloys, including microarc oxidation [5,6], organic coatings [7,8], chemical conversion coating [9,10], and so on [11–13].

Organic coatings have been widely applied to prevent metal corrosion due to their simplicity and bargain price [14–16]. Among all organic coatings, epoxy coatings are the most widely used resin matrix owing to low shrinkage, strong adhesion to substrate, and superior chemical resistance [17–19]. However, because of the volatilization of solvents during curing, epoxy coatings inevitably have micropores and defects, thus allowing corrosive media to reach the substrate through these defects and cause corrosion [20,21]. Moreover, with increase in service time, corrosive media destroy the coating structure and form corrosion channels, thus causing corrosion of the substrate [22,23]. The anticorrosion property of the epoxy coating is easily degraded over the service lifespan due to these shortcomings. Hence, improving the protection performance of epoxy coatings to satisfy the requirement of severe corrosion fields is of great importance.

Fortunately, much research has reported that nanomaterials can enhance the anticorrosion performance of coatings [24–26]. The integrity of the coating is improved because

the micropores within the coating are filled by the nanomaterials [7]. In addition, the diffusion of corrosion media is hindered by the addition of nanomaterials, delaying the arrival of corrosion ions to the substrate [27]. Nano alumina (nano-$Al_2O_3$) has been widely used in the coating protection field because of its superior hardness, chemical stability, low price, and excellent corrosion resistance [28–30]. However, nano-$Al_2O_3$ has hydrophilic surfaces decorated with hydroxyl groups, and the nanoparticles can easily adhere to each other through hydrogen bonding, leading to irregular agglomerations. Additionally, high surface energy is another reason for the agglomeration of nano-$Al_2O_3$. The agglomeration of nano-$Al_2O_3$ and the disadvantage of poor compatibility with coatings limit the application of nano-$Al_2O_3$ in coatings. Therefore, in order to obtain a nanocomposite coating with uniform nanofiller distribution, it is urgent to solve the agglomeration phenomenon of nanomaterials.

Poly-ethyleneimine (PEI) is a kind of typical Lewis base with abundant amine groups and high water solubility [31]. The special structure of this branched-chain polymer could prevent the agglomeration of nanomaterial [32]. Furthermore, the amine groups of PEI molecules can physically interact with the hydroxyl groups of nano-$Al_2O_3$ via hydrogen bonding action. The surface of the nano-$Al_2O_3$ was grafted with reactive functional groups so that they could chemically bind to the epoxy groups of the resin molecules. The interactions make the dispersion more stable, facilitating the formation of homogeneous nanocomposite. Finally, as a surface modifying agent for nano-$Al_2O_3$ in the epoxy–amine system, PEI has the advantage of improving the curing potential of epoxy nanocomposite coatings [15].

Hence, in this work, nano-$Al_2O_3$ was first functionalized with PEI and then used to prepare an $Al_2O_3$–based composite epoxy coating to solve the agglomeration phenomenon of nanomaterials. Subsequently, the anticorrosion properties of the epoxy nanocomposite coating and the corresponding anticorrosion mechanisms were investigated at length. The as-prepared epoxy nanocomposite coating exhibited favorable corrosion resistance and has broad prospects in corrosion protection.

## 2. Materials and Methods

### 2.1. Materials

Epoxy resin (H228A) and matched curing agent (H228B) were purchased from Shanghai Hanzhong Chemical Co., Ltd., Shanghai, China. Nano alumina (nano-$Al_2O_3$), sodium hydroxide (NaOH), *N*,*N*-Dimethylformamide (DMF) and ethylene imine polymer (PEI) were provided by Aladdin, Shanghai, China. All adopted chemicals and solvents were used without further purification.

### 2.2. Pretreatment of Substrate

We used 6082 aluminum alloy plates with a specification of 20 mm × 20 mm × 5 mm as the substrates. The plates were first polished with sandpapers of 400 #, 600 #, 800 #, and 1000 # to remove the surface oxidation layer. After that, the plates were sonicated in ethyl alcohol absolute three times to remove any impurities.

### 2.3. Functionalization of Nano-$Al_2O_3$

The functionalization of nano-$Al_2O_3$ was prepared by using the following procedure. Firstly, the nano-$Al_2O_3$ was gradually added into 5 mol/L NaOH under magnetic stirring for 24 h at 25 °C. Then, the nano-$Al_2O_3$/NaOH solution was washed using deionized water in the centrifugal machine, followed by the nano-$Al_2O_3$ being dried in the vacuum freeze-drier to obtain hydroxylated nano-$Al_2O_3$. After that, 5 g hydroxylated nano-$Al_2O_3$ was added into a mixture of 5 g PEI and 5 g DMF, and the mixture was magnetically stirred for 6 h at 60 °C. Subsequently, the final mixture was washed with ethyl alcohol absolute and then dried in the vacuum freeze-drier to obtain PEI functionalized nano-$Al_2O_3$, which was defined as PEI-$Al_2O_3$.

### 2.4. Preparation of Composite Coatings

The composite coatings were prepared through the following steps. Firstly, 0.45 g PEI-$Al_2O_3$ were ultrasonically dispersed in 12 g deionized water for 30 min to obtain a homogeneous suspension. Subsequently, 5 g epoxy resin and 10 g curing agents were added into the suspension. The suspension was mechanically stirred at the speed of 2000 rpm for 10 min to obtain a homogeneous slurry. The air bubbles of slurry were immediately removed using a vacuum oven at room temperature. After that, the slurry was painted on the plates using the spray method, and then the samples were cured at room temperature for 48 h and defined as PEI-$Al_2O_3$/EP coating. For comparison, pure epoxy coating (EP) and $Al_2O_3$/EP coating were also prepared by using the same method.

### 2.5. Characterization

The morphology of nano-$Al_2O_3$ was obtained using a transmission electron microscope (TEM, JEM-2100 F, JEOL, Tokyo, Japan). The surface functional groups of nano-$Al_2O_3$ and PEI-$Al_2O_3$ were analyzed via Fourier transform infrared spectroscopy (FTIR, Nicolet iN10, Thermo Fisher Scientific, Waltham, MA, USA). The fracture surface micromorphology of the coatings was observed via scanning electron microscope (SEM, JSM-7800F, JEOL, Tokyo, Japan). The thickness and adhesive strength of the coatings were measured using a coating thickness measurer and pull-off adhesion tester (DK-500, Deka Precision Measuring Instrument, Shenzhen, China), respectively.

The electrochemical corrosion behavior of the coatings was characterized using a electrochemical work station (VersaSTAT 3F, AMETEK, CA, USA) and the electrolyte was 3.5 wt.% NaCl solution. The corrosion test was performed with a traditional three-electrode system, in which the platinum electrode was as the counter electrode and a saturated calomel electrode and coated samples acted as the reference electrode and working electrode, respectively. Electrochemical impedance spectroscopy (EIS) analysis was conducted in the frequency range of $10^5$ Hz~$10^{-2}$ Hz with a sinusoidal voltage signal amplitude of 10 mV. The impedance spectra results were analyzed using Zview software (version 2). The polarization tests were implemented at a scan rate of 1 mV/s and $\pm$250 mV with respect to the open circuit potential (OCP).

## 3. Results and Discussion

### 3.1. FTIR Spectra of PEI and Nanofillers

The morphology and microstructure of nano-$Al_2O_3$ were studied via TEM as shown in Figure 1a. It is clear that the nano-$Al_2O_3$ formed spherical nanoparticles with a diameter of about 20 nm. The FTIR spectra of PEI, nano-$Al_2O_3$, and PEI-$Al_2O_3$ are shown in Figure 1b. For nano-$Al_2O_3$, the peak located at 3447 cm$^{-1}$ was due to the presence of -OH on the surface. The absorption bands at about 500~900 cm$^{-1}$ were attributed to the vibration of Al-O groups [33]. The spectrum of PEI-$Al_2O_3$ exhibited some new peaks compared with nano-$Al_2O_3$. The peaks at 2952 and 2845 cm$^{-1}$ were assigned to the vibration of C-H, and the peak at 1455 cm$^{-1}$ was assigned to the stretching vibration of C-N. These new peaks confirm that $Al_2O_3$ nanofillers were successfully modified with PEI.

### 3.2. The Thickness and Adhesion of the As-Prepared Coatings

The thicknesses of the coatings are shown in Table 1, and the thicknesses of the coatings were similar at different locations, indicating that thickness was evenly distributed. Moreover, the thicknesses of the EP, $Al_2O_3$/EP, and PEI-$Al_2O_3$/EP coating were 129.3, 123.3, and 128.3 μm, respectively, implying that the preparation of the coatings had good repeatability.

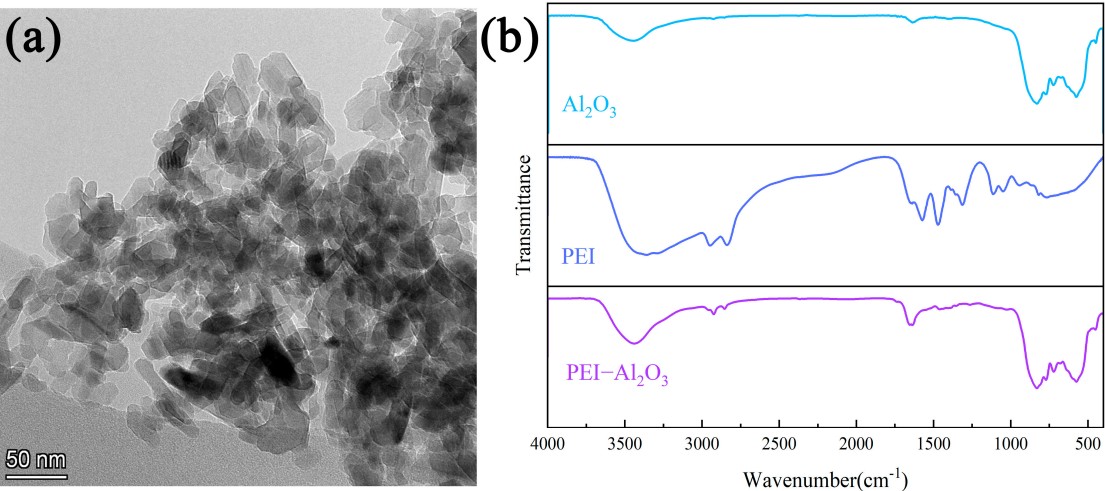

**Figure 1.** (**a**) TEM micrographs of nano−Al$_2$O$_3$ and (**b**) FTIR spectra of nano−Al$_2$O$_3$, PEI and PEI−Al$_2$O$_3$.

**Table 1.** Thickness of all as-prepared coatings.

| Sample | 1/μm | 2/μm | 3/μm | Average Value/μm |
|---|---|---|---|---|
| EP | 129 | 130 | 129 | 129.3 |
| Al$_2$O$_3$/EP | 122 | 125 | 123 | 123.3 |
| PEI-Al$_2$O$_3$/EP | 127 | 128 | 130 | 128.3 |

Figure 2 shows the adhesion strength and digital photographs of the coatings. The pull-off strengths of the EP, Al$_2$O$_3$/EP, and PEI-Al$_2$O$_3$/EP coating were 11.78, 11.98, and 12.44 MPa, respectively. This result indicates that all coatings had excellent adhesion to the substrate, which is crucial for the practical application of coatings.

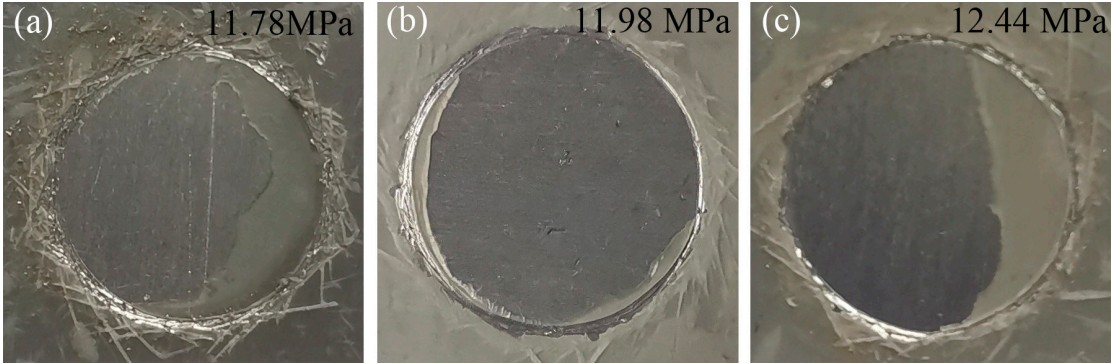

**Figure 2.** Morphology images of coatings after pull-off test: (**a**) EP coating, (**b**) Al$_2$O$_3$/EP coating, and (**c**) PEI−Al$_2$O$_3$/EP coating.

### 3.3. The Cross-Sectional Morphology of As-Prepared Coatings

Figure 3 exhibits SEM images of the cross-sections of the EP, Al$_2$O$_3$/EP, and PEI-Al$_2$O$_3$/EP coatings. The pure EP coating exhibited a smooth fracture surface while some micropores were observed within the coating, which were due to the volatilization of solvent during curing. The morphologies of the cross-sections of the Al$_2$O$_3$/EP and PEI-Al$_2$O$_3$/EP coating became rough as shown in Figure 3b,c. Additionally, nano-Al$_2$O$_3$ was detected in the rough area via EDS analysis. For the Al$_2$O$_3$/EP coating, severe aggregation of Al$_2$O$_3$ was observed in the coating. However, for the PEI-Al$_2$O$_3$/EP coating, nano-Al$_2$O$_3$ was evenly distributed within the coating compared to the Al$_2$O$_3$/EP coating.

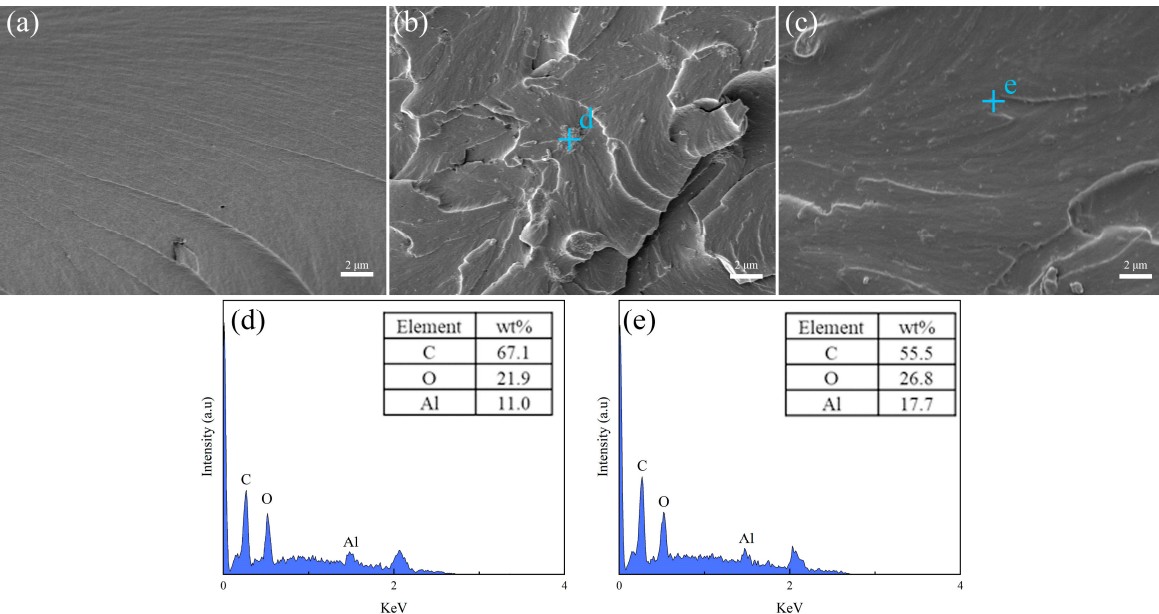

**Figure 3.** Fracture surface morphologies of coatings: (**a**) EP coating, (**b**) $Al_2O_3$/EP coating, and (**c**) PEI$-$$Al_2O_3$/EP coating. EDS point of: (**d**) d in (**b**) and (**e**) e in (**c**).

### 3.4. Electrochemical Corrosion Behavior

The corrosion behaviors of the as-prepared coatings were investigated via EIS measurement. Figure 4 shows the Nyquist and Bode plots of the coatings. With the increase in immersion time, the capacitance arc radii of all coatings in the Nyquist plots shrunk, indicating that the anticorrosion performances of all coatings gradually decreased as a result of the intrusion of corrosive media. Notably, PEI-$Al_2O_3$/EP exhibited a larger capacitive arc than that of EP and the $Al_2O_3$/EP coating during the entire immersion process, demonstrating that the PEI-$Al_2O_3$/EP coating could better protect the metal substrate. A higher lowest-frequency impedance modulus in Bode plots ($|Z|_{f=0.01\ Hz}$) usually indicates a better anticorrosion performance [34]. With the addition of nano-$Al_2O_3$, the $|Z|_{f=0.01\ Hz}$ values of the $Al_2O_3$/EP and PEI-$Al_2O_3$/EP coatings were higher than that of the pure EP coating. The $|Z|_{f=0.01\ Hz}$ value of the PEI-$Al_2O_3$/EP coating stayed the highest among all coatings during immersion. In addition, its $|Z|_{f=0.01\ Hz}$ value still stayed the highest ($1.69 \times 10^6\ \Omega\cdot cm^2$) even after 20 days of immersion, which was about two times larger than that of pure EP. These results indicate that the PEI-$Al_2O_3$/EP coating can better protect metal substrate from corrosive media.

To quantitatively analyze the corrosion resistance of the as-prepared coatings, the EIS results were further fitted with an equivalent electric circuit as shown in Figure 5a. In the equivalent electrical circuits, $R_s$, $R_c$, $Q_c$, $R_{ct}$, and $Q_{dl}$ represent solution resistance, coating resistance, coating capacitance, charge-transfer resistance, and double-layer capacitance, respectively. Generally speaking, a coating with a higher coating resistance ($R_c$) value means a preferable corrosion resistance [35]. Figure 5b shows the variation in $R_c$ values. During immersion, the $R_c$ values of all coatings exhibited a tendency to reduce, indicating that the coatings were losing their barrier effects gradually. $Al_2O_3$/EP and PEI-$Al_2O_3$/EP both exhibited larger $R_c$ values than the EP coating at the time of immersion, owing to the barrier property of nano-$Al_2O_3$. After immersing for 3 days, the $R_c$ value of the PEI-$Al_2O_3$/EP coating was $6.61 \times 10^6\ \Omega\cdot cm^2$, which is about 4 times and 10 times higher than those of the $Al_2O_3$/EP coating and EP coating, respectively. The above analysis indicates that the barrier property of the composite coating was obviously improved after adding PEI-$Al_2O_3$.

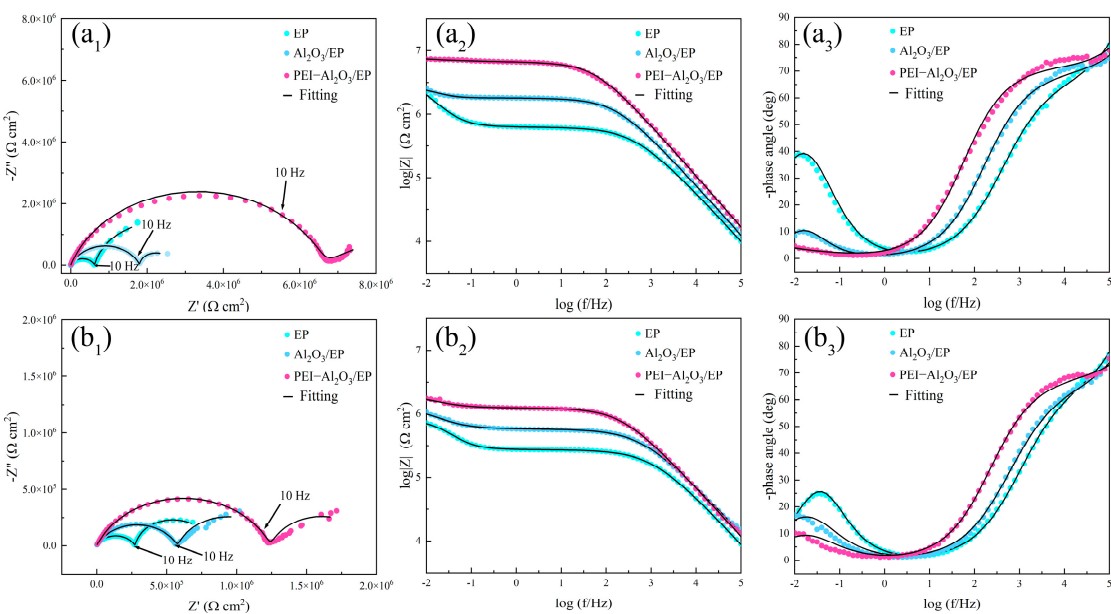

**Figure 4.** EIS results of all coatings after different immersion times: (**a₁**–**a₃**) 3 days and (**b₁**–**b₃**) 20 days.

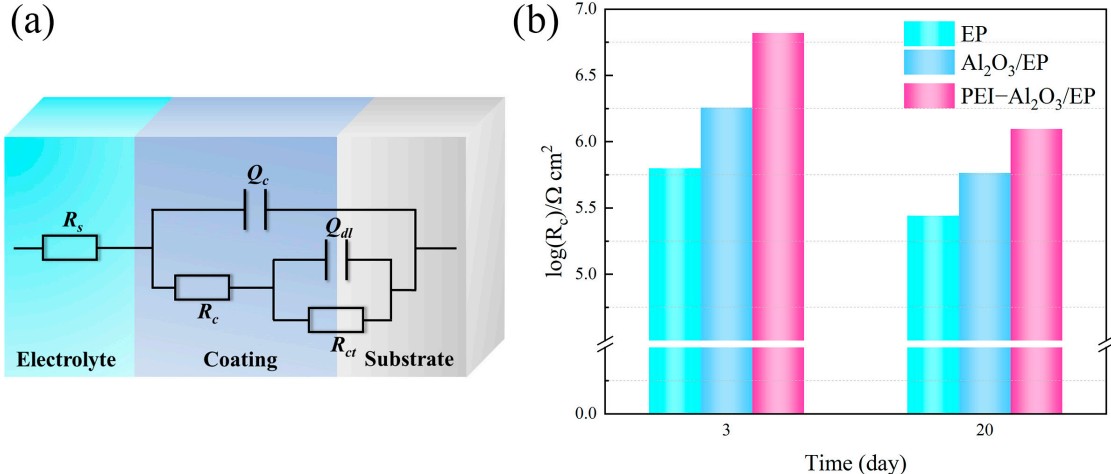

**Figure 5.** (**a**) Electrical equivalent circuit used for fitting the EIS results and (**b**) the $R_c$ values of coatings after 3 days and 20 days of immersion in 3.5 wt.% NaCl solution.

### 3.5. Potentiodynamic Polarization Test

The anticorrosion properties of the as-prepared coatings were further analyzed using a potentiodynamic polarization test. The potentiodynamic polarization curves of all coatings immersed in 3.5 wt.% NaCl solution for 12 h are shown in Figure 6. The corrosion potentials ($E_{corr}$) and corrosion current densities ($i_{corr}$) are shown in Figure 7. Typically, a more positive $E_{corr}$ and a smaller $i_{corr}$ in the polarization curve indicate a lower corrosion tendency and a lower corrosion rate, respectively [36]. The $E_{corr}$ of the PEI-Al₂O₃/EP coating ($-0.579$ V) was higher than that of the Al₂O₃/EP coating ($-0.628$ V) and the EP coating ($-0.740$ V). With the addition of nanofillers, the $i_{corr}$ of the Al₂O₃/EP and PEI-Al₂O₃/EP coatings both decreased. Moreover, the $i_{corr}$ of the PEI-Al₂O₃/EP coating ($5.388 \times 10^{-9}$ A·cm²) was lower than that of the Al₂O₃/EP coating ($2.8195 \times 10^{-8}$ A·cm²), indicating that a composite coating with homodisperse nano-Al₂O₃ exhibits better corrosion resistance.

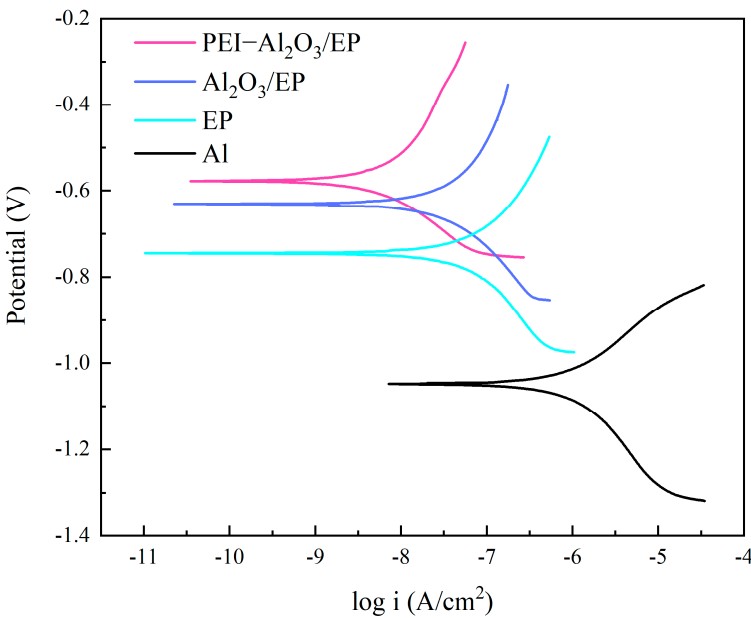

**Figure 6.** Tafel curves of samples after 12 h of immersion in 3.5 wt.% NaCl solution.

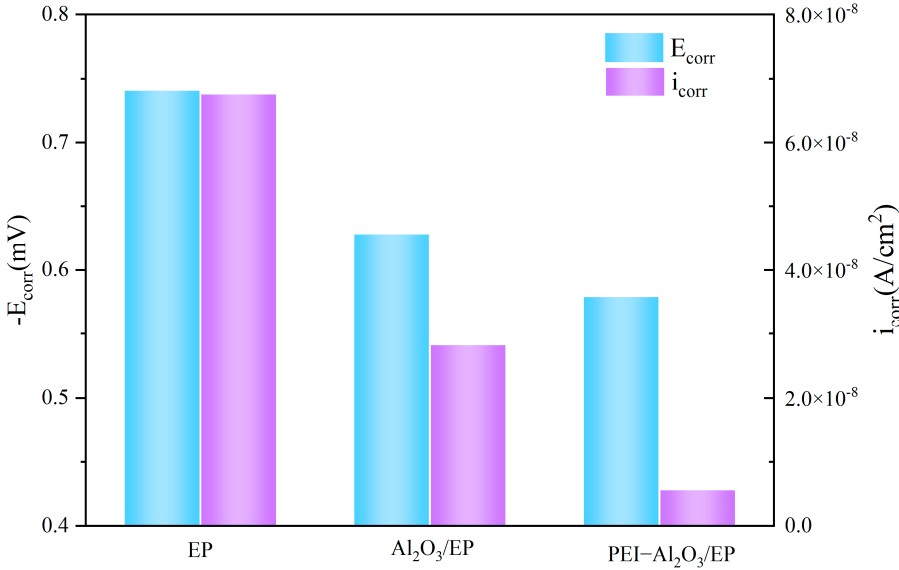

**Figure 7.** The $E_{corr}$ and $i_{corr}$ of all coatings after being immersed in 3.5 wt.% NaCl solution for 12 h.

### 3.6. Corrosion Products Analysis

The corrosion morphologies and corrosion products on coated substrates after 30 days of immersion were further investigated. Figure 8 shows the corrosion morphologies and corresponding EDS results, and the weight ratios of Al and O elements at different coating–substrate interfaces are shown in Figure 9. There were abundant corrosion products on EP-coated substrate, and the oxygen content was the highest (60.56 wt.%) compared with other samples, indicating that the EP coating had the worst corrosion resistance. For $Al_2O_3$/EP coating, the corrosion products decreased compared with the EP coating, and the oxygen content was 38.86 wt.%, implying that nano-$Al_2O_3$ could improve the anticorrosion performance of the coating by prolonging the diffusion path of corrosive media. Notably, compared to the other substrates, the surface of the PEI-$Al_2O_3$/EP substrate had the fewest corrosion products, and the trace of polishing could still be clearly observed. Moreover, the oxygen content of the PEI-$Al_2O_3$/EP substrate was 3.26 wt.%. These results suggest that the PEI-$Al_2O_3$/EP coating had the best protective effect. In addition, the corrosion products at the PEI-$Al_2O_3$/EP coating–substrate interface were further investigated via

XRD as shown in Figure 10. The XRD results show that the corrosion products at the PEI-Al$_2$O$_3$/EP coating–substrate interface mainly consisted of Al$_2$O$_3$, Al(OH)$_3$, and AlCl$_3$.

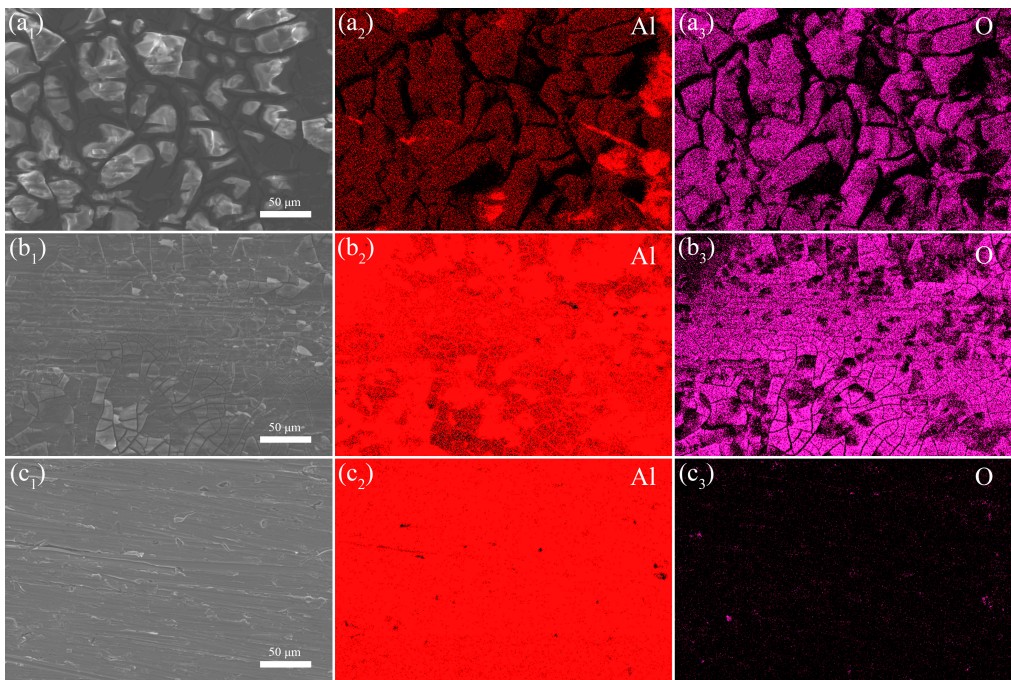

**Figure 8.** Surface morphology and element mapping of different coated substrates after immersion: (**a**) EP coating, (**b**) Al$_2$O$_3$/EP coating, and (**c**) PEI−Al$_2$O$_3$/EP coating.

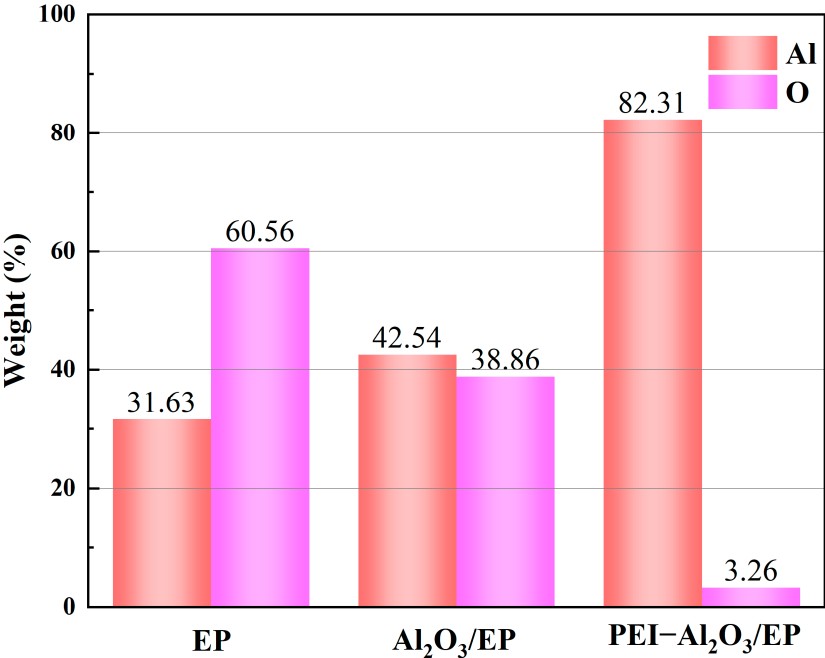

**Figure 9.** The weight ratio of Al and O in the sample surface after 20 days of immersion in 3.5 wt.% NaCl solution.

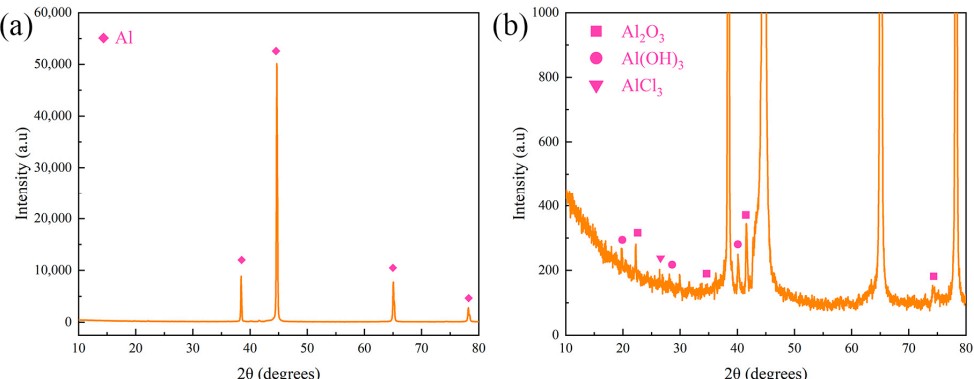

**Figure 10.** XRD pattern of corrosion products: (**a**) full figure and (**b**) local magnification figure.

### 3.7. Anticorrosion Mechanism

The anticorrosion mechanisms of the coatings are exhibited in Figure 11. The organic coating acts as an important protective layer that can protect the substrate from aggressive media. However, for the pure EP coating, there were a considerable amount of micropores within the coating, and corrosive media ($H_2O$, $O_2$, $Cl^-$) could penetrate into the coating quickly through these micropores. For the $Al_2O_3$/EP coating and the PEI-$Al_2O_3$/EP coating, the nano-$Al_2O_3$ could fill the defects in the coating and create tortuous diffusion paths, thus reducing the diffusion rate of electrolytes. However, for the $Al_2O_3$/EP coating, the agglomeration of nano-$Al_2O_3$ restricted the enhancement of its anticorrosion properties. PEI-$Al_2O_3$ with good dispersion could maximize the diffusion paths of the aggressive media within the coating. Therefore, the PEI-$Al_2O_3$/EP coating has the better anticorrosion properties.

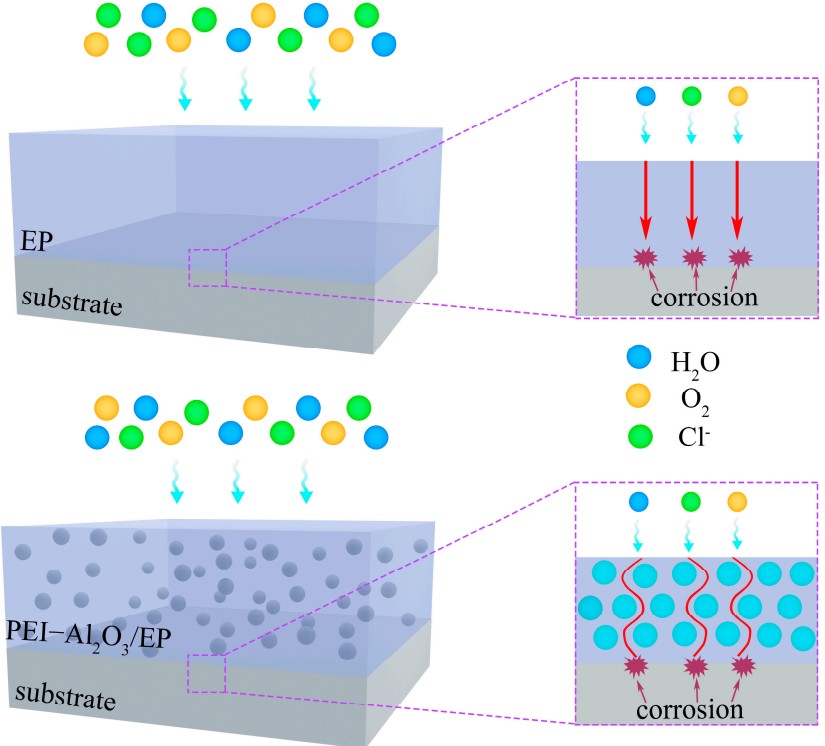

**Figure 11.** Schematic diagram of anticorrosion mechanism.

### 4. Conclusions

In summary, $Al_2O_3$ nanofillers were successfully modified with ethylene imine polymer, and an PEI-$Al_2O_3$/EP coating was successfully fabricated. Its corrosion resistance and

the corresponding mechanism were investigated in detail. The major conclusions can be drawn as follows.

(a)　The PEI-Al$_2$O$_3$ nanoparticles have excellent dispersive properties in the coating;

(b)　The coating resistance of the PEI-Al$_2$O$_3$/EP coating ($6.61 \times 10^6$ $\Omega \cdot cm^2$) was 10 times larger than that of the EP coating because nano-Al$_2$O$_3$ could fill the defects within the coating and slow the diffusion rate of corrosion media;

(c)　The surface of the PEI-Al$_2$O$_3$/EP substrate had the fewest corrosion products and lowest oxygen content (3.26 wt.%) compared to EP and the Al$_2$O$_3$/EP coating, indicating PEI-Al$_2$O$_3$/EP has the best protective effect.

**Author Contributions:** Conceptualization, S.S. and X.F.; methodology, C.H. (Cheng Hua), M.Z. and Y.Z.; software, X.L., S.S. and Y.H.; validation, X.L., S.S. and Y.H.; formal analysis, X.L. and S.S.; investigation, C.H. (Cheng Hua), M.Z. and Y.H.; resources, X.L. and X.F.; data curation, X.L. and C.H. (Can He); writing-original draft preparation, X.L., C.H. (Can He) and S.S.; writing-review and editing, X.L., C.H. (Cheng Hua), X.F., S.S. and M.C.; visualization, C.H. (Cheng Hua), M.Z. and Y.Z.; supervision, X.F.; project administration, X.F.; funding acquisition, X.F. All authors have read and agreed to the published version of the manuscript.

**Funding:** This research was carried out with the financial support provided by the National Natural Science Foundation of China (No. 52075458 and No. U2141211) and the Sichuan Science and Technology Program (No. 2021JDRC0094).

**Institutional Review Board Statement:** Not applicable.

**Informed Consent Statement:** Not applicable.

**Data Availability Statement:** Not applicable.

**Acknowledgments:** The authors gratefully acknowledge ceshigo (www.ceshigo.com, accessed on 31 December 2022) for providing testing services and the Analytical and Testing Center of Southwest Jiaotong University for supporting the SEM measurements.

**Conflicts of Interest:** The authors declare no conflict of interest.

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
