# Peer review of "Preparation of Anticorrosive Epoxy Nanocomposite Coating Modified by Polyethyleneimine Nano-Alumina"

_coatings, doi:10.3390/coatings13030561_

Round 1

Reviewer 1 Report

The present work is devoted to the study of anti corrosive composite coating on aluminium alloy. The work is interesting and well written and I can suggest publication after the following major revisions

-Please improve the introduction section considering more generally the subject of composite coatings with particles addition such as https://doi.org/10.1016/j.surfcoat.2021.127901 and https://doi.org/10.1007/BF02706445

-Please state clearly at the end of the introduction the novelty of the work

-Please describe the reproducibility of the electrochemical tests (number of samples tested for each condition)

-Please describe TEM in the experimental

-Please add OM images of the cross section of the coated specimens in order to have visual observation of the thickness, adhesion, different layers etc

-Presentation of the EIS tests should be remarkably improved, results should be fitted with a proper equivalent circuit and the results of the fitting presented. In detail in the paper is completely missing the choice of the circuit and what circuit was employed

-Please add error bars in Fig.5 and 7

Author Response

Response to reviewer’s comments

Thanks are given to the Editor and Reviewers for their insightful comments on our manuscript. These comments are very helpful for us to improve the quality of the manuscript. All the comments/suggestions have been taken into account and the revised texts are marked in red in the manuscript. We have modified the manuscript accordingly, and detailed corrections are listed below.

Reviewer 1: The present work is devoted to the study of anticorrosive composite coating on aluminium alloy. The work is interesting and well written and I can suggest publication after the following major revisions.

Question 1: Please improve the introduction section considering more generally the subject of composite coatings with particles addition such as https://doi.org/10.1016/j.surfcoat.2021.127901 and https://doi.org/10.1007/BF02706445

Answer 1: Thanks for your comments.

According to your helpful comments, we have added these papers in the manuscript. The revised part is as follows. “Thus, various strategies have been adopted to retard the corrosion rate of alloys, including microarc oxidation [5,6], organic coatings [7,8], chemical conversion coating [9,10], and so on [11-13].”

  1. Pezzato, L.; Lorenzetti, L.; Tonelli, L.; Bragaggia, G.; Dabalà, M.; Martini, C.; Brunelli, K. Effect of SiC and borosilicate glass particles on the corrosion and tribological behavior of AZ91D magnesium alloy after PEO process. Surf. Coat. Tech. 2021, 428, 127901.
  2. Agarwala, R. C.; Agarwala, V. Electroless alloy/composite coatings: A review. Sadhana, 2003, 28, 475-493.

Question 2: Please state clearly at the end of the introduction the novelty of the work.

Answer 2: Thanks for your suggestion.

According to your helpful comments, we have revised the novelty of the work in the manuscript. The revised part is as follows. “in this work, nano-Al2O3 was first functionalized with PEI and then used to solve the agglomeration phenomenon of nanomaterials.”

Question 3: Please describe the reproducibility of the electrochemical tests (number of samples tested for each condition).

Answer 3: Thank your comments.

In this work, we immerse three samples simultaneously to carry out electrochemical tests. After the three samples were tested, the stable data was selected for fitting to ensure the accuracy of the data.

Question 4: Please describe TEM in the experimental.

Answer 4: Thank you.

According to your helpful comments, we have described TEM in the experimental. The revised part is as follows. “The morphology of nano-Al2O3 is obtained by transmission electron microscope (TEM, JEOL 2100 F).”

Question 5: Please add OM images of the cross section of the coated specimens in order to have visual observation of the thickness, adhesion, different layers etc.

Answer 5: Thank your comments.

Your suggestion is meaning. To obtain the thickness and adhesion of coatings in this work, the thickness of coatings is measured by coating thickness measurer as Table R1. Meanwhile, the adhesion strength of coatings was observed by pull-off adhesion tester (DK-500, China) as shown in Fig. R1. In addition, the number of layers of coatings via spraying is all one.

Table R1. Thickness of all as-prepared coatings.

Sample

1/μm

2/μm

3/μm

average value/μm

EP

129

130

129

129.3

Al2O3/EP

122

125

123

123.3

PEI-Al2O3/EP

127

128

130

128.3

Figure R1. Morphology images of coatings after pull-off test: (a) EP coating (b) Al2O3/EP coating and (c) PEI-Al2O3/EP coating.

Question 6: Presentation of the EIS tests should be remarkably improved, results should be fitted with a proper equivalent circuit and the results of the fitting presented. In detail in the paper is completely missing the choice of the circuit and what circuit was employed.

Answer 6: Thanks for your suggestion.

Your suggestion has meaning. According to your comments, we have added the electrical equivalent circuit for fitting the EIS results as shown in Figure R2. In the equivalent electrical circuits, Rs, Rc, Qc, Rct, and Qdl represented solution resistance, coating resistance, coating capacitance, charge-transfer resistance, and double-layer capacitance, respectively.

Figure R2. Electrical equivalent circuit used for fitting the EIS results.

Question 7: Please add error bars in Fig.5 and 7.

Answer 7: Thanks for your comments.

In this work, The Rc values of samples in Fig.5 are obtained from the fitting of equivalent circuits. In the research reports, the Rc values are presented in tabular form without error bars as shown in Table R2 and Table R3. To make the comparison more striking, I have used bar charts in the manuscript. Moreover, the Ecorr and icorr of samples are obtained from Tafel curves, and the data is usually presented in tabular form without errors as shown in Table R4. In this work, the fitting chi-squared of EP, Al2O3/EP coating and PEI-Al2O3/EP coating are 0.00037, 0.00387, and 0.00358, respectively. The low fitting chi-squared represents the accuracy of the fit. This is our opinion, please point out if there are any problems, thanks again.

Table R2. Electrochemical data obtained via equivalent circuit fitting of the EIS curves. [1]

Table R3. Electrochemical parameters of bare (B) MS after immersion in 3.5 wt.% NaCl solution for 0.5 h and the evolution of electrochemical parameters of superhydrophobic SiO2/PDMS (S) coated MS over the immersion time. [2]

Table R4. The results of PDP tests for simple and composite PEO coatings on aluminum and its alloys. [3]

[1] L.Y. Cui, S.D. Gao, P.P. Li, R.C. Zeng, F. Zhang, S.Q. Li, E.H. Han, Corrosion resistance of a self-healing micro-arc oxidation/polymethyltrimethoxysilane composite coating on magnesium alloy AZ31, Corros. Sci. 118 (2017) 84-95.

[2] X.F. Zhang, Y.Q. Chen, J.M. Hu, Robust superhydrophobic SiO2/polydimethylsiloxane films coated on mild steel for corrosion protection, Corros. Sci. 166 (2020) 108452.

[3] K. Babaei, A. Fattah-alhosseini, M. Molaei, The effects of carbon-based additives on corrosion and wear properties of Plasma electrolytic oxidation (PEO) coatings applied on Aluminum and its alloys: A review, Surf. Interfaces 21 (2020) 100677.

Thank you very much for the excellent reviewing work and helpful comments. We are looking forward to receiving your positive message.

Reviewer 2 Report

Manuscript ID coatings-2262878 entitled:

Preparation of Anticorrosive Epoxy Nanocomposite Coating Modified by Polyethyleneimine Nano-alumina

Authors

Xin Liang , Cheng Hua , Mingrui Zhang , Yu Zheng , Shijie Song , Meng Cai , Yu Huang , Can He , Xiaoqiang Fan *

General comment

This manuscript reports the results regarding anticorrosion properties of 6082 aluminum alloy plates covered with    nano-alumina (nano-Al2O3) modified with polyethyleneimine (PEI) epoxy.

Some  recommendations and observation are listed below:

1. What is the number of replicates for each test? 

2. In R 183, the authors mention that the modeling of the data was done according to an equivalent electrical circuit, but they do not describe the circuit used and the meaning of the circuit elements chosen. A proper confirmation of the equivalent circuit and its individual elements with the physico-chemical surface state is missing. Nyquist plots should point out some frequencies. A more  detailed discussion is beneficial.

3. The presence of two capacitative semicircles are observed especially at prolonged immersion time. The measured impedance spectra could be explained in terms of the general model for the description of the electrochemical behaviour of a metal covered with a film with pores. In this case in the equivalent circuit consists  the resistance of the electrolyte, the capacitance  of the  coating layer, the pore resistance due to penetration of electrolyte, the charge transfer resistance, and the double-layer capacitance at the substrate/electrolyte interface are present. All fitted parameters (CPE, R, n, ……) should be given and discussed.

4.  At R187 the authors write “…the Al2O3/EP and PEI-Al2O3/EP both exhibited larger Rc values than EP coating in the time of immersion, attributing to the barrier property of nano-Al2O3. Check the text, the mentioned resistance value does not correspond to the one shown in fig 5 for 3 days of immersion, but rather after 20 days of immersion.

5. Insert in figure 6 the Tafel curves for the uncovered metal substrate.

6. Check the icorr values ​​presented in the text. The intersection of the Tafel slopes (see figure) shown in figure 6 indicates different values.

Author Response

Response to reviewer’s comments

Thanks are given to the Editor and Reviewers for their insightful comments on our manuscript. These comments are very helpful for us to improve the quality of the manuscript. All the comments/suggestions have been taken into account and the revised texts are marked in red in the manuscript. We have modified the manuscript accordingly, and detailed corrections are listed below.

Reviewer 2: This manuscript reports the results regarding anticorrosion properties of 6082 aluminum alloy plates covered with nano-alumina (nano-Al2O3) modified with polyethyleneimine (PEI) epoxy.

Question 1: What is the number of replicates for each test?

Answer 1: Thank you very much.

In order to make the experimental data reliable, three replicate experiments were carried out. In addition, after the three samples were tested, the stable data was selected for fitting to ensure the accuracy of the data.

Question 2: In R 183, the authors mention that the modeling of the data was done according to an equivalent electrical circuit, but they do not describe the circuit used and the meaning of the circuit elements chosen. A proper confirmation of the equivalent circuit and its individual elements with the physico-chemical surface state is missing. Nyquist plots should point out some frequencies. A more  detailed discussion is beneficial.

Answer 2: Thank you.

According to your helpful comment, we have added A more detailed discussion. We have added the electrical equivalent circuit for fitting the EIS results as shown in Figure R1. In the equivalent electrical circuits, Rs, Rc, Qc, Rct, and Qdl represented solution resistance, coating resistance, coating capacitance, charge-transfer resistance, and double-layer capacitance, respectively. Moreover, we have pointed out some frequencies in the Nyquist plots according to your suggestion as shown in Figure R2.

Figure R1. Electrical equivalent circuit used for fitting the EIS results.

Figure R2. EIS results of all coatings after different immersion time: (a1-a3) 3 days and (b1-b3) 20 days.

Question 3: The presence of two capacitative semicircles are observed especially at prolonged immersion time. The measured impedance spectra could be explained in terms of the general model for the description of the electrochemical behaviour of a metal covered with a film with pores. In this case in the equivalent circuit consists  the resistance of the electrolyte, the capacitance  of the  coating layer, the pore resistance due to penetration of electrolyte, the charge transfer resistance, and the double-layer capacitance at the substrate/electrolyte interface are present. All fitted parameters (CPE, R, n, ……) should be given and discussed.

Answer 3: Thank you for pointing these out to us.

Your comments are helpful. According to your comments of equivalent circuit, we have added the equivalent circuit consists solution resistance, coating resistance, coating capacitance, charge-transfer resistance, and double-layer capacitance as shown in Figure R3. In the equivalent electrical circuits, Rs, Rc, Qc, Rct, and Qdl represented solution resistance, coating resistance, coating capacitance, charge-transfer resistance, and double-layer capacitance, respectively. Generally speaking, a coating with higher coating resistance (Rc) value means a preferable corrosion resistance. During immersion, the Rc values all coatings showed a tendency to reduce, indicating that the coatings were losing its barrier effect gradually. The Al2O3/EP and PEI-Al2O3/EP both exhibited larger Rc values than EP coating in the time of immersion, attributing to the barrier property of nano-Al2O3. Moreover, the anticorrosion properties of as-prepared coatings were further analyzed by potentiodynamic polarization test. The above characterizations give indications of the corrosion protection properties of the coating. So other fitted parameters (CPE, R, n, ……) are not discussed in this paper

Figure R3. Electrical equivalent circuit used for fitting the EIS results.

Question 4: At R187 the authors write “…the Al2O3/EP and PEI-Al2O3/EP both exhibited larger Rc values than EP coating in the time of immersion, attributing to the barrier property of nano-Al2O3. Check the text, the mentioned resistance value does not correspond to the one shown in fig 5 for 3 days of immersion, but rather after 20 days of immersion.

Answer 4: Thank you.

According to your comments, we have checked the mentioned resistance value. The Rc values of Al2O3/EP and PEI-Al2O3/EP are larger than the EP coatings for the whole immersion time. The Rc values of PEI-Al2O3/EP coating after 3 days and 20 days are 6.61 × 106 Ω·cmand 1.24 × 106 Ω·cm2, respectively.

Question 5: Insert in figure 6 the Tafel curves for the uncovered metal substrate.

Answer 5: Thank you.

According to your helpful comments, we have added Tafel curves of the uncovered metal substrate. The revised figure is shown in figure R4.

Figure R3. Tafel curves of samples after 12 h of immersion in 3.5 wt.% NaCl solution.

Question 6: Check the icorr values presented in the text. The intersection of the Tafel slopes (see figure) shown in figure 6 indicates different values.

Answer 6: Thanks for your suggestions.

According to your helpful suggestion, we have checked the icorr values in the manuscript. The revised part is “the icorr of PEI-Al2O3/EP coating (5.388×10-9 A·cm2) was lower than that of Al2O3/EP coating (2.8195×10-8 A·cm2), indicating that composite coating with homodisperse nano-Al2O3 exhibits better corrosion resistance.” Thank you again for your guidance.

Thank you very much for the excellent reviewing work and helpful comments. We are looking forward to receiving your positive message.

Round 2

Reviewer 1 Report

The paper is now suitable for publication

Reviewer 2 Report

Accept in the present form.